# Transcriptomic and Hormonal Changes in Wheat Roots Enhance Growth under Moderate Soil Drying

**DOI:** 10.3390/ijms25179157

**Published:** 2024-08-23

**Authors:** Ying Li, Shuqiu Jiang, Yonghui Hong, Zixuan Yao, Yadi Chen, Min Zhu, Jinfeng Ding, Chunyan Li, Xinkai Zhu, Weifeng Xu, Wenshan Guo, Nanyan Zhu, Jianhua Zhang

**Affiliations:** 1Jiangsu Key Laboratory of Crop Genomics and Physiology/Jiangsu Key Laboratory of Crop Cultivation and Physiology, College of Agriculture, Yangzhou University, Yangzhou 225009, China; yingli@yzu.edu.cn (Y.L.); dx120230131@stu.yzu.edu.cn (S.J.); mz120211245@stu.yzu.edu.cn (Y.H.); 006514@yzu.edu.cn (M.Z.); jfdin@yzu.edu.cn (J.D.); licy@yzu.edu.cn (C.L.); xkzhu@yzu.edu.cn (X.Z.); guows@yzu.edu.cn (W.G.); 2Department of Biology, Hong Kong Baptist University, Hong Kong 999077, China; 3Jiangsu Co-Innovation Center for Modern Production Technology of Grain Crops, Yangzhou University, Yangzhou 225009, China; 4College of Horticulture and Landscape Architecture, Yangzhou University, Yangzhou 225009, China; yadichen@yzu.edu.cn; 5Joint International Research Laboratory of Water and Nutrient in Crop, College of JunCao Science and Ecology, Fujian Agriculture and Forestry University, Fuzhou 350002, China; wfxu@fafu.edu.cn; 6School of Life Sciences and State Key Laboratory of Agrobiotechnology, The Chinese University of Hong Kong, Hong Kong 999077, China

**Keywords:** plant hormone, metabolism, soil drying, wheat, root

## Abstract

Understanding the mechanisms that regulate plant root growth under soil drying is an important challenge in root biology. We observed that moderate soil drying promotes wheat root growth. To understand whether metabolic and hormonic changes are involved in this regulation, we performed transcriptome sequencing on wheat roots under well-watered and moderate soil drying conditions. The genes upregulated in wheat roots under soil drying were mainly involved in starch and sucrose metabolism and benzoxazinoid biosynthesis. Various plant hormone-related genes were differentially expressed during soil drying. Quantification of the plant hormones under these conditions showed that the concentrations of abscisic acid (ABA), cis-zeatin (CZ), and indole-3-acetic acid (IAA) significantly increased during soil drying, whereas the concentrations of salicylic (SA), jasmonic (JA), and glycosylated salicylic (SAG) acids significantly decreased. Correlation analysis of total root length and phytohormones indicated that CZ, ABA, and IAA are positively associated with wheat root length. These results suggest that changes in metabolic pathways and plant hormones caused by moderate soil drying help wheat roots grow into deeper soil layers.

## 1. Introduction

Drought limits crop growth and yield worldwide [1,2,3]. Modulation of root systems is a key feature of plant responses to soil drying [4]. Rhizosphere soils form a complex environment, and roots subjected to soil drying face multiple challenges that can alter their growth and development [5]. Maintenance of root growth is essential for plant adaptation to soil drying. However, the mechanisms by which soil drying mediates developmental changes are unclear.

Wheat (*Triticum aestivum* L.) is important for global food security, but its production is severely reduced by drought in many regions [3]. Researchers have found that a 40% reduction in water can cause a 20.6% yield loss in wheat, a threat that is exacerbated by climatic warming and population expansion [6,7,8]. Beneficial plant root traits and irrigation management techniques that can enhance water-use capacities and increase crop yields under reduced water input need to be identified urgently [9]. Moderate soil drying is a promising approach for promoting root growth, reducing water inputs, and maintaining or increasing crop yields.

Plant hormones such as abscisic acid (ABA), auxin, brassinosteroid, cytokinin, ethylene, gibberellic acid (GA), jasmonic acid (JA), salicylic acid (SA), and strigolactone are important plant growth regulators that play pivotal roles in responses to various environmental stresses [10,11,12,13]. Classic studies have shown that ABA biosynthesis and concentrations increase under drought conditions, and low levels of applied ABA or osmotic stress can increase root growth [4]. Drought stress causes ABA to accumulate in plant roots, enter the xylem, and be transported to the guard cells of leaves via transpiration, inducing stomatal closure, which reduces water transpiration under drought stress [14,15]. Drought stress can induce the expression of genes involved in the water deficit response in roots, such as pathogenesis-related protein 4, ABA deficient protein 4, ABA insensitive protein 5, protein phosphatase 2C, and ABA-8′ hydroxylase [10,16]. The first plant hormone discovered, auxin, regulates virtually all aspects of plant growth and development, as well as adaptation to drought, high temperature, and salinity [1,5,17]. The auxin-responsive *Gretchen Hagen 3* (*GH3*) family gene *WES1* (for WESO 1, meaning a dwarfed stature in Korean), which encodes indole-3-acetic acid (IAA) amino acid synthase, is up-regulated under stress conditions, catalyzing the binding to IAA of amino acids, causing inactivation and thereby reducing endogenous auxin levels, and activating the expression of the stress-related genes *pathogenesis-related protein 1* (*PR-1*) and *C-repeat binding factor* (*CBF*) to adapt to stress [18]. Other plant hormones also play important roles under drought conditions; for instance, perturbation of ethylene or cytokinin pathways can affect growth and survival under drought conditions [11,12,13]. In addition to plant hormones, several metabolic pathways regulate the adaptive responses of plant roots to soil drying [19,20,21,22,23,24,25,26].

In nature, plant roots grow in a complex soil environment with extremely diverse soil microbial communities, and studies suggest that rhizosphere microorganisms in the soil significantly affect root growth and resistance to various stresses [5,27]. For example, auxin-producing bacterial strains isolated from soil (*Chryseobacterium culicis* and *Paenibacillus polymyxa*) produce IAA and enhance barley root growth [27]. However, it is not clear how plant hormone actions and metabolic pathways are integrated into plant root growth and stress adaptation under dry soil conditions.

We hypothesized that metabolic pathways and phytohormones play important roles in mediating root growth during soil drying. We analyzed differentially expressed genes (DEGs) in wheat roots between well-watered (WW) and soil-drying (SD) conditions using transcriptome sequencing and examined the concentrations of various phytohormones under soil drying. Our results suggest that starch and sucrose metabolism and benzoxazinoid biosynthesis promote wheat root elongation under moderate soil drying, allowing roots to obtain more water from the soil.

## 2. Results

### 2.1. Moderate Soil Drying Promotes Wheat Root Growth

Our previous study indicated that moderate drought can promote root growth in *Arabidopsis* and tomatoes, allowing the plants to obtain more water [28]. To investigate the effect of soil drying on wheat root growth, wheat was grown under well-watered (WW) and soil-drying (SD) conditions. Under WW, soil water was maintained at 80% of the field capacity for the first 4 days and then dried for 2 days, whereas in SD, soil water was maintained at 45% of the field capacity for the first 4 days before drying for 2 days. We found that moderate soil drying significantly promoted wheat root growth (Figure 1A). Compared to WW, the root length and fresh and dry weights per plant under SD were 19.1%, 29.4%, and 53.3% higher, respectively (Figure 1B–D). In addition, there was no significant difference in shoot fresh weight (Appendix A) or dry weight (Appendix A) between WW and SD. Taken together, moderate soil drying can significantly promote root growth.

### 2.2. Starch and Sucrose as Primary Metabolites and Benzoxazinoids as Secondary Metabolites Are Involved in Wheat Root Growth Under Soil Drying

To investigate the mechanism by which moderate soil drying promotes wheat root growth, we collected roots under WW and SD conditions for RNA-Seq analysis. After filtering the raw data, checking the sequencing error rate, and checking the GC content distribution, clean reads (Appendix A) for subsequent RNA-Seq analysis were obtained. The expression of all genes in each sample was calculated, and the distribution of gene expression levels in different samples was evaluated (Appendix A). The violin plot for each sample displays five statistics (from top to bottom, the maximum value, upper quartile, median, lower quartile, and minimum value); this shows that the data distribution in each sample is concentrated, which ensures the accuracy and reliability of the RNA-Seq analysis in this experiment. Pearson correlation analysis showed that the square of the correlation coefficient (R2) between each pair of samples exceeded 0.82 (Figure 2A), and principal component analysis (PCA) revealed that the six samples could be clearly assigned to the WW and SD groups (Figure 2B), indicating that the data from each independent biological replicate in this experiment had good reproducibility and could be used for further analysis. The gene expression of each sample was analyzed; 49,567 genes were detected in the WW group (3078 unique genes) versus 49,378 genes in the SD group (2889 unique genes) (Figure 2C). Differential gene clustering analysis showed that many genes were differentially expressed in the different conditions (Appendix A). Compared to WW, SD treatment significantly upregulated (more than 2-fold) 1958 genes and downregulated (less than 2-fold) 2529 genes (Figure 2D and Appendix A), indicating that the expression of many genes in wheat roots changed after soil drying.

To identify which functional classes or biological processes the DEGs are involved in, we performed Gene Ontology (GO) functional enrichment analysis. GO is a comprehensive database that describes gene functions, including biological processes (BP), cellular components (CC), and molecular functions (MF). We selected the ten processes with the greatest significant differences from each functional category for analysis, totaling 30 biological processes involving response to water deprivation, response to water, structural consistency of chromatin, vacuole, and other processes (Figure 3). These results indicated that many genes responsive to water deficiency are involved in wheat root elongation under soil drying.

Next, we performed a Kyoto Encyclopaedia of Genes and Genomes (KEGG) pathway enrichment analysis on the DEGs. Figure 4 shows the ten KEGG pathways with the greatest differences; the DEGs were mainly enriched in metabolic pathways. Compared to WW, the KEGG pathways with more upregulated genes under SD treatment included biotin, linoleic acid, cyanoamino acid, starch, and sucrose metabolism, as well as the biosynthesis of benzoxazinoids and various plant secondary metabolites. Of these, SD treatment had the most upregulated genes for the biosynthesis of various plant secondary metabolites and starch and sucrose metabolism. The upregulated genes in the secondary metabolite pathway were mainly enriched in benzoxazinoid biosynthesis (Figure 5). Collectively, these results suggest that starch, sucrose, and benzoxazinoids contribute to wheat root elongation under soil drying.

### 2.3. Soil Drying Increases the CZ, ABA, and IAA Levels in Wheat Roots

Studies have indicated that phytohormones are involved in root growth under drought, so we next analyzed plant hormone-regulated DEGs. Compared with WW, 29 auxin-related genes were differentially expressed in wheat roots under SD treatment (19 significantly upregulated, ten significantly downregulated) (Figure 6A); 13 ABA-related genes showed differential expression (five significantly upregulated, eight significantly downregulated) (Figure 6B); four jasmonic acid (JA)-related genes were significantly downregulated (Figure 6C); three cytokinin-related genes showed differential expression (two significantly upregulated, one significantly downregulated) (Figure 6D); and three salicylic acid (SA)-related genes were significantly downregulated (Figure 6E). To verify the reliability of the transcriptome data, we randomly selected plant hormone-related genes for qPCR detection. The results (Figure 7) showed that qPCR data had a high correlation with the data obtained from the RNA-Seq (Figure 6), with a correlation ranging from 0.82 to 0.91, indicating that the RNA-Seq results were reliable (*p* < 0.05). To further examine the role of plant hormones in wheat roots under different water treatments, we measured the concentrations of 34 phytohormones (Appendix A) in wheat roots in eight major categories: auxin, abscisic acid, cytokinins, melatonin, jasmonic acid, ethylene, salicylic acid, and gibberellins. Only 16 plant hormones were detected and quantified (Figure 8), and the other 18 plant hormones were not detected due to their low content in samples. Compared to WW, concentrations of cis-zeatin (CZ) (Figure 8C), abscisic acid (ABA) (Figure 8J), and indole-3-acetic acid (IAA) (Figure 8M) were significantly increased under SD treatment. Four plant hormones were significantly decreased: JA (Figure 8F), SA (Figure 8K), glycosylated salicylic acid (SAG) (Figure 8L), and indole-3-formaldehyde (ICAId) (Figure 8O). Correlation analysis of total root length and plant hormones showed that CZ, ABA, and IAA are positively correlated with root length (Figure 9). These results indicate that CZ, ABA, and IAA are significantly increased under moderate soil drying, which may contribute to wheat root elongation under soil drying.

### 2.4. Analysis of the Interactions of Plant Hormone Levels and Metabolic Pathways

To analyze the relationships between hormone-coupled metabolic pathways, we performed correlation analysis of the KEGG metabolic pathway analysis on the plant hormones. As shown in Figure 10, ABA and JA affected the biosynthesis of plant secondary metabolites; SA, IAA, IP, ABA, JA, and Ile affected the plant hormone signal transduction pathway; and CZ, SA, IAA, ABA, JA, and ethylene affected the biosynthesis of plant hormones pathway. Among these plant hormones, the log2Fold Change value of ABA is the largest and affects almost all metabolic pathways. These results suggest that moderate soil drying affects plant hormone signaling transduction and plant hormone synthesis pathways, especially altering the ABA and IAA concentrations, ultimately promoting wheat root growth to obtain more water from the soil.

## 3. Discussion

Under mild soil drying, root elongation is maintained or even promoted [29,30]. In this study, our results indicate that moderate soil drying promotes wheat root growth (Figure 1), which is consistent with results in *Arabidopsis* and tomato [10,28]. Wheat root transcriptome analysis indicates that the genes upregulated under soil drying were mainly enriched in starch and sucrose metabolism and benzoxazinoid biosynthesis (Figure 2, Figure 3, Figure 4 and Figure 5). Root growth is closely related to soil and rhizosphere microorganisms. Studies have found that sugar concentrations in maize root exudates affect the rhizosphere bacterial community [25]. Sugars are derived from plant photosynthesis, are transported by the phloem to various places, and also play an important role in the regulation of plant growth and development. Sugars have been found in abundance in root exudate compounds and have been suggested to affect the root-associated microbiomes [25]. Chaparro et al. found that the abundances of *Acidobacteria*, *Actinobacteria*, *Bacteroidetes*, and *Cyanobacteria* were correlated with the levels of root exudated amino acids, phenols, sugars, and alcohols [22]. Cordovez et al. found that sugars in tomato root exudates attract *Pseudomonas* to colonize them and help them resist fungi [26]. In addition, roots can shape a specific rhizosphere microbial community structure by secreting various secondary metabolites [24]. Benzoxazinoids have a screening effect on rhizosphere microorganisms, and the tolerance of bacteria to benzoxazinoids shapes the maize rhizosphere microbiome [23]. Therefore, soil drying may affect the secretion of these metabolites by altering the expression of genes related to sucrose and benzoxazinoids, reshaping the microbial structure of the rhizosphere soil, thereby affecting wheat root growth.

Several plant hormones regulate root growth and stress adaptation responses [15,17,18]. Auxin accumulation and gradient modulation by multiple auxin transporters play crucial roles in the regulation of plant growth and development [18,27,31,32,33]. ABA is one of the main plant hormones involved in water deficiency and the induction of stomatal closure [14,15,17]. Under moderate water stress, ABA establishes and maintains root meristem function and stimulates root elongation [34,35,36]. ABA response and auxin transport play a key role in root elongation under drought stress; for example, auxin transporter mutants *aux1-7* and *eir1-4* (*pin2*) and ABA synthesis mutant *aba3-1* exhibit significantly lower root elongation rates than the wild type under osmotic stress conditions [1]. We found that IAA- and ABA-related genes are involved in wheat root growth (Figure 6 and Figure 7). The hormone quantification results showed that moderate drought stress increases the IAA and ABA levels (Figure 8), consistent with our previous finding that moderate drought increases the IAA and ABA concentrations in tomato and *Arabidopsis* roots [10,28]. The correlation analysis of the metabolic pathway analysis of plant hormones showed that ABA also affects the biosynthesis of secondary metabolites (Figure 10). Previous studies have shown that lower auxin concentrations lead to a reduction in the meristem size of the root tip and reduced root growth [17]. ABA accumulation restricts ethylene synthesis in roots under water stress conditions, and ethylene can also interact with auxin in plant roots [1,17,30]. In addition, we found that moderate drought increases plant CZ levels [21], which is consistent with our results (Figure 8 and Figure 9). Therefore, an increase in CZ concentrations may benefit wheat root growth during soil drying.

Plant hormones not only affect the root structure [19,20] but are also secreted outside the roots, affecting the soil rhizosphere microbial community to regulate plant growth [25,37]. Some previous studies characterized changes in the rhizosphere microbiome due to added plant hormones or changes in phytohormone signal transduction in roots. For instance, immune-signaling phytohormone SA mutants and added SA were employed to indicate that SA dramatically regulates colonization of the root microbiome by specific bacterial taxa [38]. JA is another important phytohormone involved in defense signaling that has been suggested to shape the rhizosphere microbiome. JA and sugars are important root exudates that affect the composition of the maize rhizobacterial communities, and the soil within the bacterial community within rows is modulated by different JA concentrations [25]. Changes in SA signals can alter the relative abundance of specific bacterial communities, such as actinomycetes in the rhizosphere microbial community [37]. In addition, hormonal crosstalk networks suggest various layers of complexity in the regulation of root growth by water stress conditions; for example, the responses of phytohormone synthesis, transport, and signal transduction components to water stress are complex and nonlinear, and understanding the effects of one phytohormone requires comprehensive consideration of how this phytohormone affects all other components [17]. Our results indicate that moderate soil drying reduced the JA and SA concentrations (Figure 8 and Figure 9), perhaps due to the soil conditions. In summary, all of these results suggest that moderate soil drying affects plant metabolic and hormone pathways or the rhizosphere microbial community, ultimately promoting wheat root growth to obtain more water from deep soil. Understanding the mechanisms regulating root elongation under soil drying is pivotal to our understanding of the rooting patterns of plants under natural conditions, as efforts intensify to improve the water-use efficiency traits of crops in order to meet the demand for food under increasingly climate-challenged growth conditions.

## 4. Materials and Methods

### 4.1. Plant Materials and Growth Conditions

This study used the wheat (*Triticum aestivum* L.) variety ‘Fielder’. Wheat seeds with full grains and uniform size were selected before treatment. Wheat seeds were surface-sterilized using 10% (*v*/*v*) sodium hypochlorite for 15 min and rinsed five times with double-distilled water [39]. The seeds were subsequently placed on moistened filter paper in darkness for 36 h at 28 °C to synchronize germination. Then, the device was placed in a greenhouse with a 14 h light (under 100 mmol photons m^−2^ s^−1^ illumination) and a 10 h dark cycle, a 26 °C light and a 22 °C dark temperature cycle, and a relative humidity of 60%.

### 4.2. Soil Drying Treatment

The soil used in this study was collected from a wheat field in Yangzhou, Jiangsu Province, China (119°53′ E, 32°31′ N), from a depth of 0–20 cm, and air dried for 7 days. Soil chemical factors are shown in Appendix A. The air-dried soil was sieved through a 4 mm mesh to remove any coarse material and vegetative matter. Then, 20% sterile distilled water by weight was added to the soil, and the mixture was mixed thoroughly and passed through a 4 mm mesh again to ensure uniform soil particle size. For subsequent water addition from the bottom to help the water absorb naturally and prevent the soil from hardening, nine small holes of uniform size were created in the bottoms of the black pots (9 cm upper diameter, 6 cm lower diameter, 17 cm height). Then, the sieved soil (500 g) was placed in the black pots. Subsequently, uniform seedlings whose roots had grown to a length of about 2 cm at 36 h after germination were cultivated. The soil-water treatment was performed as previously described by Xu et al. [27], with slight modifications. Different amounts of water were added to the bottom trays of the pots, and the wheat plants were subjected to two water treatments: well-watered conditions (WW, maintaining 80% of field capacity for the first 4 days, and then drying for 2 days) and moderate soil drying (SD, maintaining 45% of field capacity for the first 4 days, and then drying for 2 days).

### 4.3. Root Sample Collection and Analysis

Wheat roots were carefully collected and shaken after the pots were disassembled. The roots sampled from the washed plants were weighed to obtain the fresh weight and placed together in an oven (40 °C for 7 days) to obtain the dry weight. The total root length was measured using ImageJ software (Version 1.54d).

### 4.4. Transcriptome Sequencing Process and Quantitative Real-Time PCR (qRT-PCR) Analysis

The surfaces of wheat roots were washed clean, and then the roots were quickly frozen in liquid nitrogen. The root samples were sent via cold chain to Nuohe Zhiyuan Technology Service (Beijing, China) for total RNA extraction using a RNAprep Pure Plant Plus Kit (DP441; TIANGEN, Beijing, China) according to the manufacturer’s instructions and sequencing using an Agilent 2100 bioanalyzer to evaluate the RNA quality and quantity. After passing inspection, different libraries were pooled according to the effective concentration and target offline data volume requirements for Illumina sequencing. Reads with adapters, poly-N, or low quality were removed to obtain clean data. All subsequent analyses are based on high-quality analysis using clean data. Aegilops tauschii (Ats) was used as the reference genome. The high-quality reads were annotated using HISAT2 (ver. 2.0.5). Gene expression, DEGs, and gene function were analyzed on the Nuohe Zhiyuan Cloud platform, and charts were drawn on the website https://www.chiplot.online/ (accesed on 24 January 2024).

Total RNA was extracted from roots using a RNAprep Pure Plant Plus Kit (DP441; TIANGEN, China) according to the manufacturer’s instructions. First-strand cDNA was synthesized using a First-Strand cDNA Synthesis Kit (TransGen Biotech, Beijing, China) according to the manufacturer’s instructions. Quantitative real-time PCR (qRT-PCR) assays were performed as described by Li et al. [39]. The expression levels were normalized to the expression of the wheat *Actin-6* gene. The primers specific to each gene are listed in Appendix A.

### 4.5. Root Plant Hormone Determination

After soil drying treatment, wash the surface of the wheat roots with 1 × PBS (130 mM NaCl, 7 mM Na_2_HPO_4_·7H_2_O, 3 mM NaH_2_PO_4_·4H_2_O, pH 7.0), dry it with sterile filter paper, cut the wheat roots into sterile foil with sterile scissors, and quickly place it in liquid nitrogen for freezing. At least six seedlings were pooled for each biological replicate for hormonomic analysis, with three biological replicates for each sample. The root samples were sent via cold chain to PANOMIX Biomedical Tech (Suzhou, China) for plant hormone detection using liquid chromatography-mass spectrometry (LC-MS/MS). UPLC separation was performed on an ExionLC UPLC system (AB Sciex, Boston, MA, USA) equipped with an Acquity UPLC^®^ CSH C18 (1.7 μm, 2.1 mm × 150 mm, Waters, Milford, MA, USA) column. The temperature of the column was set at 40 °C. The sample injection volume was 2 μL. The eluents consisted of 0.05% formic acid with 2 mM ammonium formate water (eluent A) and 0.05% formic acid methanol (eluent B). The MS analysis was performed using an AB Sciex Triple Quadrupole 6500 Plus mass spectrometer (AB Sciex, USA) in the multiple reaction monitoring (MRM) mode. Appendix A shows the fitting of the standard curves of plant hormones in wheat roots under WW and SD conditions.

### 4.6. Statistical Analysis

All experimental data were analyzed using SPSS version 17.0. Each treatment was repeated at least three times, and the differences between the two groups were examined using the Student’s *t*-test; Duncan’s test was used to analyze the differences between two or more datasets [1,39]. *p* < 0.05 was considered a significant difference.

## 5. Conclusions

This study comprehensively elucidated the responding mechanisms of soil drying on wheat root elongation based on physiological-biochemical performance and transcriptional analysis. RNA-Seq analysis indicated that the upregulated genes in the wheat roots under SD were mainly enriched in starch and sucrose metabolism pathways and the biosynthesis pathway of benzoxazinoids. Various plant hormone-related genes were also differentially expressed under SD. The concentrations of CZ, ABA, and IAA increased significantly under SD, whereas the SA, JA, SAG, and indole-3-formaldehyde (ICADd) concentrations significantly decreased under SD. ABA, JA, and ethylene affect the biosynthesis of plant secondary metabolites under SD. These results suggest that changes in these metabolic pathways and plant hormones contribute to wheat root elongation to obtain water from deeper layers. In conclusion, this study investigated transcriptomes and hormones in wheat roots under soil drying stress and will be useful for clarifying the mechanism of drought stress in wheat.

## Figures and Tables

**Figure 1 ijms-25-09157-f001:**
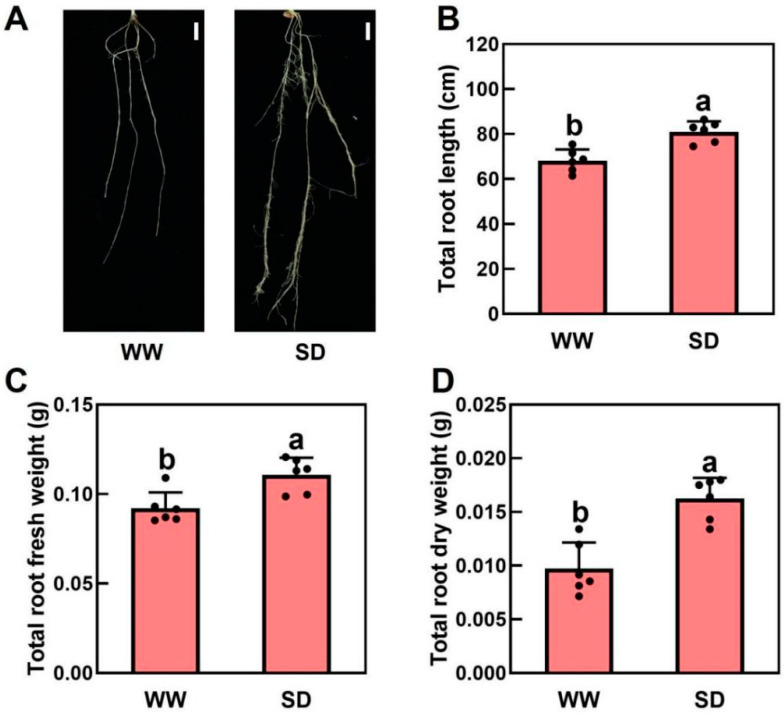
Effects of soil drying on wheat roots. (**A**) Representative images of wheat roots in well-watered (WW) and soil drying (SD) conditions. Scale bar = 1 cm. (**B**–**D**) Impact of soil drying on the total root length (**B**), root fresh weight (**C**), and root dry weight (**D**). The data in (**B**–**D**) are the mean ± standard error (n = 6). The same letter indicates no significant difference, whereas different letters indicate significant differences (*p* < 0.05, Duncan’s test).

**Figure 2 ijms-25-09157-f002:**
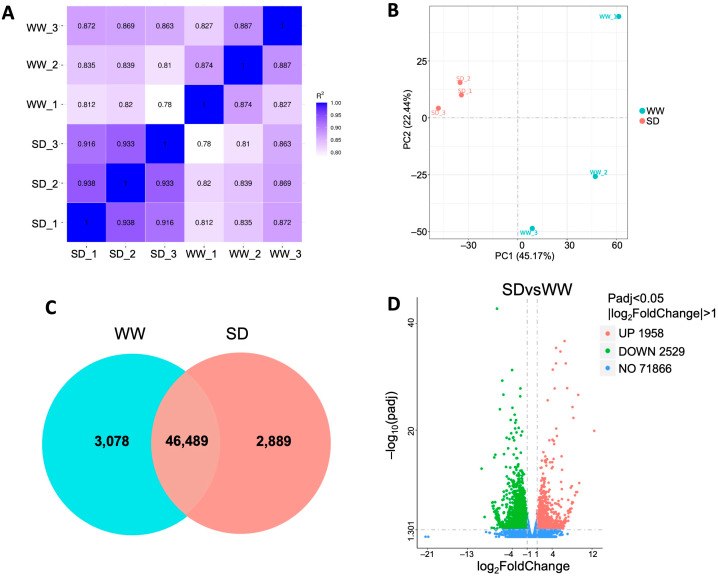
DEGs in wheat roots between well-watered (WW) and soil-drying (SD) conditions. (**A**) Pearson correlation coefficient of gene expression; (**B**) Principal components analysis of the transcriptome; (**C**) Venn diagram of DEGs; and (**D**) number of DEGs in wheat roots between the WW and SD conditions. Each point in D represents a gene, with red representing upregulated (greater than 2-fold), green representing downregulated (less than 2-fold), and blue representing genes whose expression did not change between WW and SD.

**Figure 3 ijms-25-09157-f003:**
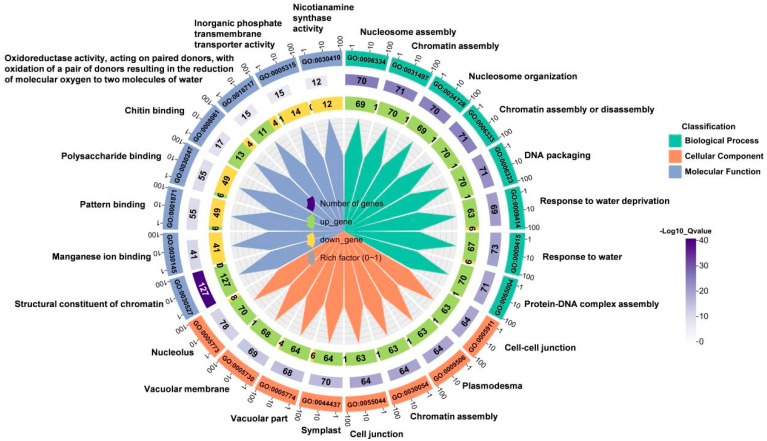
GO biological process enrichment chart of DEGs between well-watered (WW) and soil-drying (SD) conditions. The outermost circle of the image represents the term name enriched with DEGs. From outer to inner, the first layer is the GO term number, layer 2 is the number of genes in the GO term, and layer 3 shows the numbers of up- and down-regulated genes in the GO term. The color range from white to purple represents the magnitude of significance, and the gray label indicates the proportion of differentially expressed genes in the enriched pathway to the total enriched genes.

**Figure 4 ijms-25-09157-f004:**
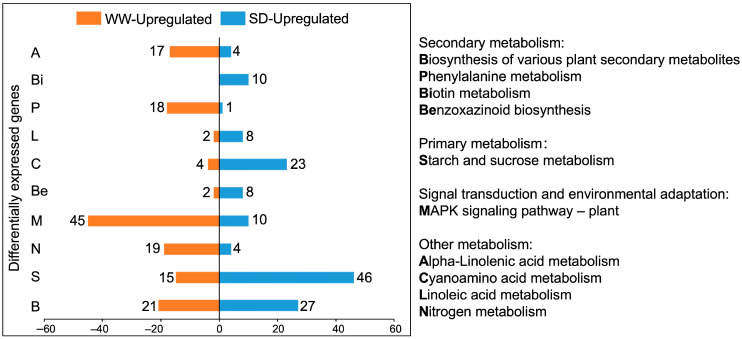
KEGG pathway enrichment chart of DEGs between well-watered (WW) and soil-drying (SD) conditions. Orange (blue) represents the number of upregulated genes under WW (SD) conditions.

**Figure 5 ijms-25-09157-f005:**
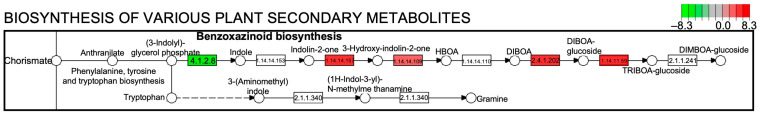
Benzoxazinoid biosynthesis in the KEGG pathway between well-watered (WW) and soil-drying (SD) conditions. The color and number in the upper right box of the metabolic pathway map represent the up- and down-regulated gene expression levels in the pathway.

**Figure 6 ijms-25-09157-f006:**
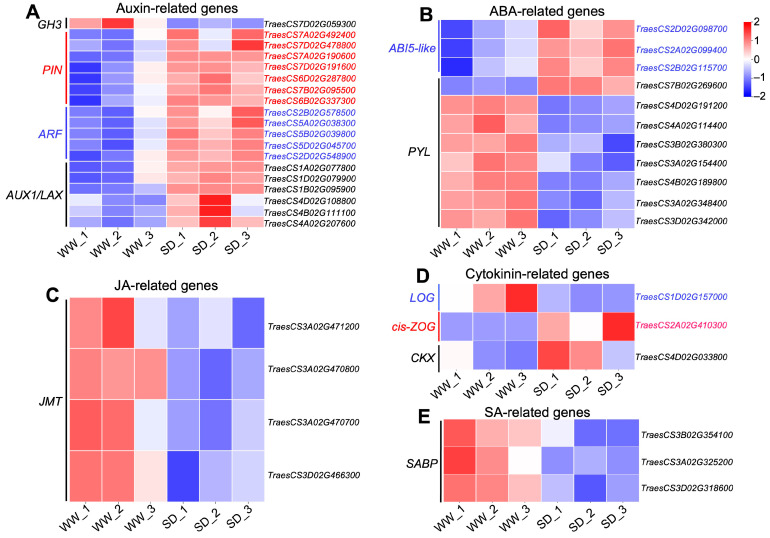
Heatmap of DEGs related to phytohormones between well-watered (WW) and soil-drying (SD) conditions. Heat map of auxin-related (**A**), ABA-related (**B**), JA-related (**C**), cytokinin-related (**D**), and SA-related (**E**) DEGs between WW and SD. The color of the heat map represents the expression of genes, with blue (red) indicating low (high) gene expression.

**Figure 7 ijms-25-09157-f007:**
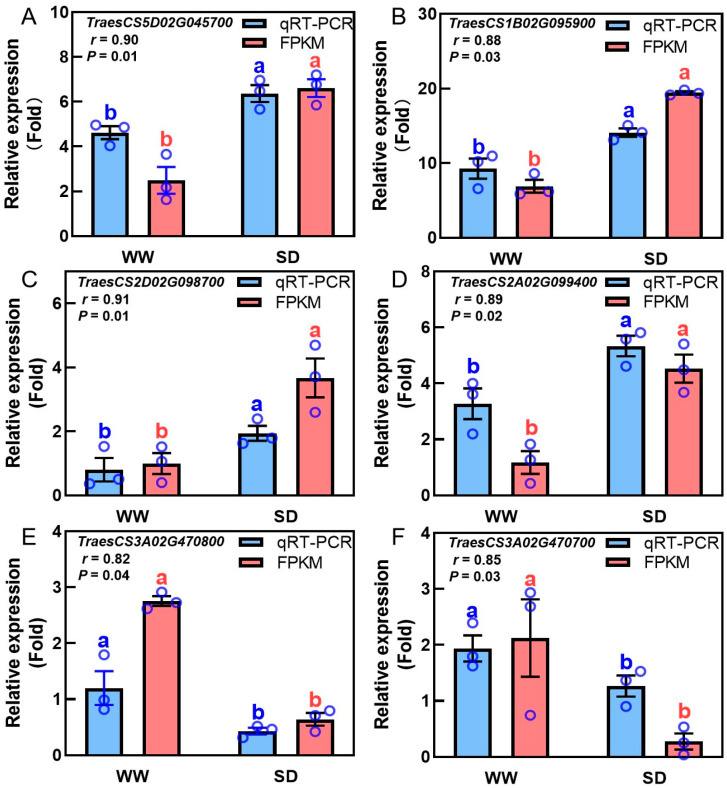
Correlation analysis between transcriptome profiling and qRT-PCR verification results of phytohormone-related genes in wheat roots under well-watered (WW) and soil-drying (SD) conditions. Auxin—(**A**,**B**), ABA—(**C**,**D**), and JA-related (**E**,**F**) gene expression in wheat roots under WW and SD conditions. The data in (**A**–**F**) are presented using mean ± standard error (n = 3). FPKM: the number of transcripts per million mapped reads per thousand bases of transcription. The same letter indicates no significant difference, while different letters indicate significant differences (*p* < 0.05, Student’s *t*-test).

**Figure 8 ijms-25-09157-f008:**
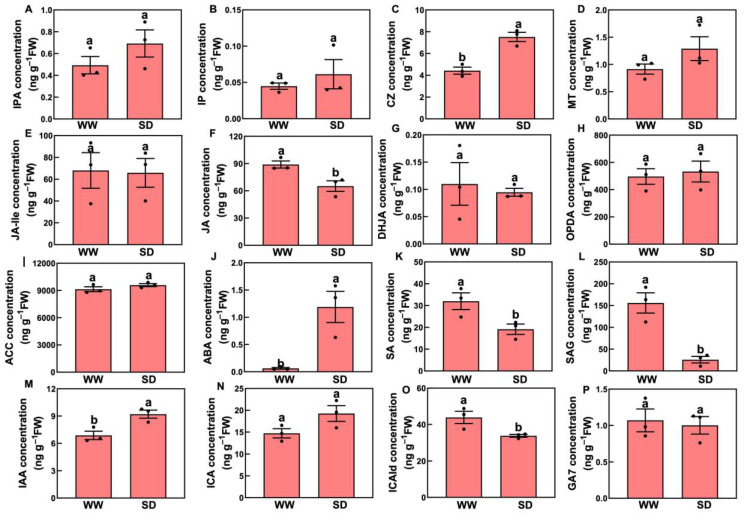
Changes in phytohormones in roots between well-watered (WW) and soil-drying (SD) conditions. (**A**–**C**) Changes in cytokinins, including isopentenyladenine nucleoside (IPA), isopentenyladenine (IP), and cis-zeatin (CZ), (**D**) Changes in melatonin. (**E**–**H**) Changes in jasmonic acids, including N-jasmonic acid isoleucine (JA-Ile), jasmonic acid (JA), dihydrojasmonic acid (DHJA), and 12-oxo plant dienoic acid (OPDA), (**I**–**J**) Changes in 1-aminocyclopropane-1-carboxylic acid (ACC, ethylene) and abscisic acid. (**K**–**L**) Changes in salicylic acids, including salicylic acid (SA) and glycosylated salicylic acid (SAG), (**M**–**O**). Changes in auxins, including indole-3-acetic acid (IAA), 3-indole-3-carboxylic acid (ICA), and indole-3-formaldehyde (ICAId), (**P**) Changes in gibberellin 7 (gibberellins). The data in the figure is presented using mean ± standard error (n = 3). The same letter indicates no significant difference, while different letters indicate significant differences (*p* < 0.05, Student’s *t*-test).

**Figure 9 ijms-25-09157-f009:**
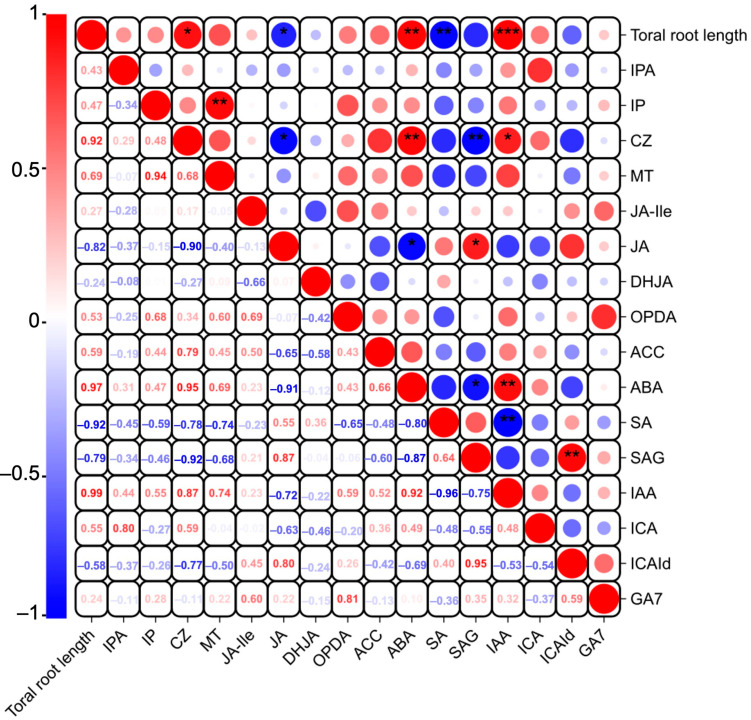
Correlation analysis of total root length and plant hormones in wheat roots between well-watered (WW) and soil-drying (SD) conditions. Asterisk indicates a significant correlation: * represents *p* < 0.05, ** represent *p* < 0.01, and *** represent *p* < 0.001. Red indicates positive correlations, while blue indicates negative ones. The size of the circle is positively correlated with the Pearson correlation coefficient.

**Figure 10 ijms-25-09157-f010:**
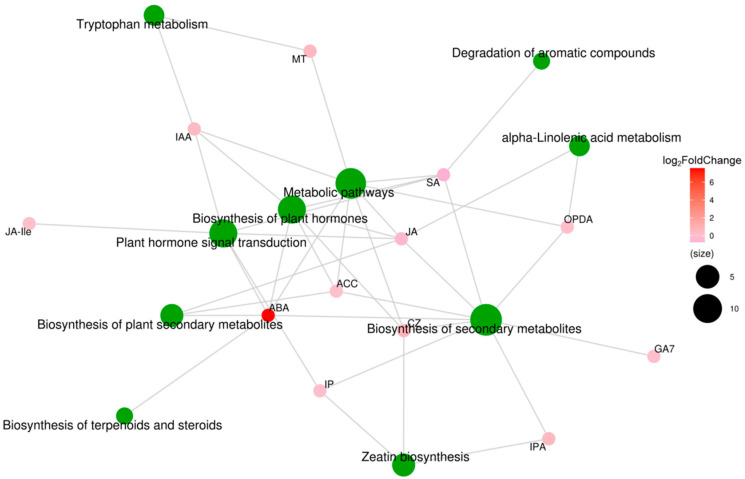
Metabolite molecular network of differentially expressed plant hormones in wheat roots between well-watered (WW) and soil-drying (SD) conditions. Green dots represent metabolic pathways, and other dots represent metabolite molecules. The size of each metabolic pathway point represents the number of metabolite molecules connected to it; the greater the number, the larger the point. Metabolite molecular points indicate the size of log_2_Fold Change values through gradual color changes, and multiple groups with no metabolite log_2_Fold Change information are compared.

## Data Availability

All data that support the findings of this study are available from the authors upon reasonable request.

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
