# Peer review of "Transcriptomic and Hormonal Changes in Wheat Roots Enhance Growth under Moderate Soil Drying"

_ijms, 2024, doi:10.3390/ijms25179157_

Round 1

Reviewer 1 Report

Comments and Suggestions for Authors

The work " Metabolic and Hormonic Changes in Wheat Root Enhance Its Growth under Moderate Soil Drying " by Li et al. (ijms-314577) aims at the effects of moderate soil on the wheat root grwoth and its transcriptomic and hormocin changes. However, there are several issues that prevent me from recommending the publication of the manuscript.

First of all, The title needs to be revised, manuscript aims at the effects of moderate soil on the wheat root grwoth and its transcriptomic and hormocin changes.

Secondly, note that the tenses in the manuscript are incorrect in several places in the introduction and discussion.

Thirdly, there are many abbreviations in the manuscript, so the full name should be written when it appears first time, and some abbreviations are not annotated in the manuscript.

Other mistakes as follow:

-line 60: ‘WES1’ should be in italics.

-line 95-97: These sentences should be placed in the main text.

-line 177: What is the unit of measurement of qPCR in the Fig.7 ? How to verify the consistence between qPCR and RNA-Seq ? It should be add the corelation between them.

Author Response

  • Responses to comments from the Reviewer #1:

The work “Metabolic and Hormonic Changes in Wheat Root Enhance Its Growth under Moderate Soil Drying” by Li et al. (ijms-314577) aims at the effects of moderate soil on the wheat root growth and its transcriptomic and hormocin changes. However, there are several issues that prevent me from recommending the publication of the manuscript.

Question #1: First of all, The title needs to be revised, manuscript aims at the effects of moderate soil on the wheat root growth and its transcriptomic and hormocin changes.

Answer: Many thanks for your good comments. We have revised the title “Transcriptomic and Hormonic Changes in Wheat Root Enhance Its Growth under Moderate Soil Drying”. Please see the revised manuscript for details (Lines 2-3).

Question #2: Secondly, note that the tenses in the manuscript are incorrect in several places in the introduction and discussion.

Answer: Many thanks for your good comments. We have revised the tenses in the introduction and discussion in our revised manuscript. Please see the revised manuscript for details (Line 73; Line 248; Line 308).

Question #3: Thirdly, there are many abbreviations in the manuscript, so the full name should be written when it appears first time, and some abbreviations are not annotated in the manuscript.

Answer: Yes, we have added the full name of abbreviations and annotated abbreviations in our revised manuscript. For example: “The auxin-responsive Gretchen Hagen 3 (GH3) family gene WES1 (for WESO 1, meaning a dwarfed stature in Korean), which encodes indole-3-acetic acid (IAA) amino acid synthase, is up-regulated under stress conditions, catalysing the binding to IAA of amino acids, causing inactivation and thereby reducing endogenous auxin levels and activating the expression of the stress-related genes pathogenesis-related protein 1 (PR-1) and C-repeat binding factor (CBF) to adapt to stress”. Please see the revised manuscript for details (Lines 62-67; Line 176, Line 179; Lines 189-193).

Question #4: line 60: ‘WES1’ should be in italics.

Answer: Yes, we have done as suggested. Please see the revised manuscript for details (Lines 62-67 ).

Question #5: line 95-97: These sentences should be placed in the main text.

Answer: Yes, we have done as suggested. Please see the revised manuscript for details (Lines 91-94).

Question #6: line 177: What is the unit of measurement of qPCR in the Fig.7 ? How to verify the consistence between qPCR and RNA-Seq? It should be add the correlation between them.

Answer: Many thanks for your good comments. The unit of measurement of qPCR is relative expression (fold), we have added the in Methods and Figure 7: “The expression levels were normalized to the expression of the wheat Actin-6 gene”. Please see the revised manuscript for details (Figure 7; Lines 368-369).

We have added the correlation between qPCR and RNA-Seq (please see the Figure 7 below), and The results (Figure 7) showed that qPCR data had a high correlation with the data obtained from RNA-Seq results (Figure 6), with a correlation ranging from 0.82 to 0.91, indicating that the RNA-Seq results were reliable (P < 0.05). Please see the revised manuscript for details (Figure 7; Lines 181-184; Lines 203-210).

Figure 7. Correlation analysis between transcriptome profiling and qRT-PCR verification results of phytohormone-related genes in wheat roots under well-watered (WW) and soil-drying (SD) conditions. Auxin- (A-B), ABA- (C-D), and JA-related (E-F) gene expression in wheat roots under WW and SD conditions. The data in the figures A-F are presented using mean ± standard error (n=3). FPKM: the number of transcripts per million mapped reads per thousand bases of transcription. The same letter indicates no significant difference, while different letters indicate significant differences (p < 0.05, Student's t-test).  

Please see the revised manuscript for details (Figure 7; Lines 181-184; Lines 368-369; Lines 203-210).

Thanks again to the referee for your careful reviewing.

With Best Wishes,

Yours truly,

Jianhua Zhang and Nanyan Zhu

The Chinese University of Hong Kong, Hong Kong, China, E-mail: jzhang@hkbu.edu.hk;

College of Animal Science and Technology, Yangzhou University, Yangzhou, China, E-mail:dx120200148@stu.yzu.edu.cn.

Reviewer 2 Report

Comments and Suggestions for Authors

In this manuscript submitted to IJMS, the authors study the effect of soil drying on hormonal and metabolic changes in the wheat root. Unfortunately, I have to say that in the manuscript are several flaws (methods are in many points incomplete and important information is missing) which prevent my accepting and major revisions are required.

1.     The „Materials and Methods“ section should be revised to provide adequate information: it is not stated how the plant material was collected for hormonomic analysis, how many replicates were performed, whether the samples were purified in any way prior to analysis.

2.     Chapter 4.5 does not provide the reader with relevant information on the LC/MS analysis. .......

3.     In Fig. S4, standard curves for a total of 34 phytohormones are presented, and it should be properly explained in the text why only 16 substances were quantified as a result (see Fig. 8).

4.     Abbreviations for individual phytohormones should be explained (not shown in the legend to Fig. S4).

5.     In my opinion, the discussion should be extended and the results should be discussed in a broader context (not only in the context of your previous publications).

Author Response

Responses to comments from the Reviewer #2:

Question #1: In this manuscript submitted to IJMS, the authors study the effect of soil drying on hormonal and metabolic changes in the wheat root. Unfortunately, I have to say that in the manuscript are several flaws (methods are in many points incomplete and important information is missing) which prevent my accepting and major revisions are required. The “Materials and Methods” section should be revised to provide adequate information: it is not stated how the plant material was collected for hormonomic analysis, how many replicates were performed, whether the samples were purified in any way prior to analysis.

Answer: Many thanks for your good comments. Yes, we have added the relevant information in the “Materials and Methods” section: “After soil drying treatment, wash the surface of the wheat roots with 1×PBS (130 mM NaCl, 7 mM Na2HPO4.7H2O, 3 mM NaH2PO4.4H2O, pH 7.0), dry it with sterile filter paper, cut the wheat roots into sterile foil with sterile scissors, and quickly place it in liquid nitrogen for freezing. At least 6 seedlings were pooled for each biological replicate for hormonomic analysis, with three biological replicates for each sample” (Lines 372-376). Please see the revised manuscript for details (Lines 372-376).

Question #2: Chapter 4.5 does not provide the reader with relevant information on the LC/MS analysis. .......

Answer: Yes, we have added the relevant information in Chapter 4.5: “The root samples were sent via cold chain to PANOMIX Biomedical Tech (Suzhou, China) for plant hormone detection using liquid chromatography-mass spectrometry (LC-MS/MS). UPLC separation was performed on ExionLC UPLC system (AB Sciex, USA) equipped with an Acquity UPLC® CSH C18 (1.7 μm, 2.1x150 mm, Waters) column. The temperature of the column was set at 40 °C. The sample injection volume was 2 μL. Eluents was consisted of 0.05% formic acid with 2 mM ammonium formate water (eluent A) and 0.05% formic acid methanol (eluent B).The MS analysis was performed using an AB Sciex Triple Quadrupole 6500 plus mass spectrometer (AB Sciex, USA) in the multiple reaction monitoring (MRM) mode” (Lines 379-385). Please see the revised manuscript for details (Lines 379-385).

Question #3:  In Fig. S4, standard curves for a total of 34 phytohormones are presented, and it should be properly explained in the text why only 16 substances were quantified as a result (see Fig. 8).

Answer: Yes, we have added the explanation why only 16 substances were quantified as a result in our revised manuscript: “To further examine the role of plant hormones in wheat roots under different water treatments, we measured the concentrations of 34 phytohormones (Figure S4) in wheat roots in eight major categories: auxin, abscisic acid, cytokinins, melatonin, jasmonic acid, ethylene, salicylic acid, and gibberellins. Only 16 plant hormones were detected and quantified (Figure 8), and the other 18 plant hormones were not detected due to their low content in samples” (Lines 184-189). Please see the revised manuscript for details (Lines 184-189).

Question #4: Abbreviations for individual phytohormones should be explained (not shown in the legend to Fig. S4).

Answer: Yes, we have added the explanation of abbreviations in the legend to Fig. S4. “Gibberellin A (GA), Abscisic acid (ABA), Salicylic acid (SA), salicylic acid 2-O-β-D- glucose (SAG), Jasmonate acid (JA), 3-Indoleformic acid (ICA), Indole-3-acetic acid (IAA), 3-Indolebutyric acid (IBA), Cis-zeatin (CZ), Trans-zeatin riboside (TZR), N6-(delta 2-Isopentenyl)-adenine (IP), N6-isopentenyladenosine-D6 (IPA), DL-dihydrozeatin (DZ), Benzyladenine (6-BA), 1-Aminocyclopropane-1-carboxylic acid (ACC), Dihydrojasmonicacid (DHJA), N-[(-)-jasmonoyl]-(S)-isoleucine (JA-Ile), Methyl 2-(1H-indol-3-yl)acetate (Me-IAA), Indole-3-carboxaldehyde (ICAId), Trans-Zeatin (TZ), Methyl jasmonate (MEJA), Brassinolide (BR), Melatonin (MT), Kinetin (Kt), 12-oxophytodienoic acid (OPDA)” (Figure S4 legend). Please see the revised manuscript for details (Figure S4 legend).

Question #5: In my opinion, the discussion should be extended and the results should be discussed in a broader context (not only in the context of your previous publications).

Answer: Many thanks for your good comments. Yes, we have added the new reference and extended the discussions, and discussed the results in a broader context:

“ Sugars are derived from plant photosynthesis and are transported by the phloem to various, and also play an important role in the regulation of plant growth and development. Sugars have been found in abundance in root exudate compounds and have been suggested to affect the root-associated microbiomes [25] ”(Lines 254-258). 

“Auxin accumulation and gradient modulated by multiple auxin transporter play crucial roles in the regulation of plant growth and development [18,27,31-33]. ABA is one of the main plant hormones involved in water deficiency and the induction of stomatal closure [14,15,17]. Under moderate water stress, ABA establishes and maintains root meristem function and stimulates root elongation [34-36]. ABA response and auxin transport play a key role in root elongation under drought stress, for example, auxin transporter mutants aux1-7 and eir1-4 (pin2) and ABA synthesis mutant aba3-1 exhibit significantly lower root elongation rate that wild type under osmotic stress conditions [1]”. (Lines 270-278).

“Previous studies have shown that lower auxin concentrations lead to a reduction in meristem size of root tip and reduced root growth [17]. ABA accumulation restricts ethylene synthesis in roots under water stress conditions, and that ethylene can also interact with auxin in plant root [1,17,30]”. (Lines 284-287)

  “Some previous studies characterized changes in the rhizosphere microbiome due to added plant hormones or changes in the phytohormones signal transduction in roots. For instance, immune-signaling phytohormone SA mutants and added SA were employed to indicate that SA dramatically regulates colonization of the root microbiome by specific bacterial taxa [38]. JA is another important phytohormone involved in defense signaling that has been suggested to shape the rhizosphere microbiome. JA and sugars are important root exudates that affects the composition of the maize rhizobacterial communities, and the soil within rows bacterial community is modulated by different JA concentrations [25]. Changes in SA signals can alter the relative abundance of specific bacterial communities, such as actinomycetes in the rhizosphere microbial community [37]. In addition, hormonal crosstalk networks suggests vaiouus layers of complexity in the regulation of root growth by water stress conditions, for example, the responses of phytohormones synthesis, transport, and signal transduction components to water stress are complex and nonlinear, and understanding the effects of one phytohormone requires comprehensively consideration of how this phytohormone affects all other components [17]. Our results indicate that moderate soil drying reduced the JA and SA concentrations (Figures 8 and 9), perhaps due to the soil conditions” (Lines 293-310).

We have cited the new reference in Discussion: “38. Lebeis, S.L.; Paredes, S.H.; Lundberg, D.S.; Breakfield, N.; Gehring, J.; McDonald, M.; Malfatti, S.; del Rio, T.G.; Jones, C.D.; Tringe, S.G.; Dangl, J.L. Salicylic acid modulates colonization of the root microbiome by specific bacterial taxa. Science. 2015, 349, 860–864. https://doi.org/10.1126/science.aaa8764”. (Lines 518-520).

Please see the revised manuscript for details (Lines 254-258; Lines 270-278; Lines 284-287; Lines 293-310; Lines 518-520).

Reviewer 3 Report

Comments and Suggestions for Authors

Dear authors, I do not understand the intention of placing the methodology after the results. I think it should be placed before and, above all, explain your decisions regarding the statistical tests presented. Furthermore, the methodology lacks some textual clarity. Explain, for example, if the concentration of sodium hypochlorite and the exposure time were based on previous studies; if an additional selection of seeds was carried out to ensure uniformity before treatment. Regarding the soil, could you present the physico-chemical characteristics, for example: texture, pH, and organic content, as this information is crucial for the reproducibility of the experiments? The reasons for sieving the soil as well as the quantities of water and frequency. Regarding the RNA extraction protocol, I believe adding details about the method would help to better understand it.

I would like to see a conclusion with a pointer to future investigations.

Best regards

Author Response

Responses to comments from the Reviewer #3:

Question #1: Dear authors, I do not understand the intention of placing the methodology after the results. I think it should be placed before and, above all, explain your decisions regarding the statistical tests presented. Furthermore, the methodology lacks some textual clarity. Explain, for example, if the concentration of sodium hypochlorite and the exposure time were based on previous studies; if an additional selection of seeds was carried out to ensure uniformity before treatment.

Answer: Many thanks for your good comments. Based on the format of International Journal of Molecular Sciences, we placed the methodology after the results. Statistical tests were performed as previously described by Xu et al. [1] and Li et al. [38] (Line 391). We have added the methodology about sample collection, RNA extraction protocol, and plant hormone determination: “ Wheat seeds with full grains and uniform size were selected before treatment. Wheat seeds were surface-sterilized using 10% (v/v) sodium hypochlorite for 15 minutes and rinsed five times with double-distilled water [39]. ”(Lines 319-322).

We have added relevant information for hormonomic analysis:

4.5. Root plant hormone determination   

  After soil drying treatment, wash the surface of the wheat roots with 1×PBS (130 mM NaCl, 7 mM Na2HPO4.7H2O, 3 mM NaH2PO4.4H2O, pH 7.0), dry it with sterile filter paper, cut the wheat roots into sterile foil with sterile scissors, and quickly place it in liquid nitrogen for freezing. At least 6 seedlings were pooled for each biological replicate for hormonomic analysis, with three biological replicates for each sample. The root samples were sent via cold chain to PANOMIX Biomedical Tech (Suzhou, China) for plant hormone detection using liquid chromatography-mass spectrometry (LC-MS/MS). UPLC separation was performed on ExionLC UPLC system (AB Sciex, USA) equipped with an Acquity UPLC® CSH C18 (1.7 μm, 2.1x150 mm, Waters) column. The temperature of the column was set at 40 °C. The sample injection volume was 2 μL. Eluents was consisted of 0.05% formic acid with 2 mM ammonium formate water (eluent A) and 0.05% formic acid methanol (eluent B). The MS analysis was performed using an AB Sciex Triple Quadrupole 6500 plus mass spectrometer (AB Sciex, USA) in the multiple reaction monitoring (MRM) mode. Figure S4 shows the fitting of the standard curves of plant hormones in wheat roots under WW and SD conditions”. (Lines 371-386).

Please see the revised manuscript for details (Lines 319-322; Lines 371-386; Line 391).

Question #2: Regarding the soil, could you present the physico-chemical characteristics, for example: texture, pH, and organic content, as this information is crucial for the reproducibility of the experiments? The reasons for sieving the soil as well as the quantities of water and frequency.

Answer: Many thanks for your good comments. We have added the physico-chemical characteristics in the revised manuscript (please see the Table S2 below). We have revised the: “The soil used in this study was collected from a wheat field in Yangzhou, Jiangsu Province, China (119°53′ E, 32°31′ N), from a depth of 0-20 cm, and air dried for 7 days. Soil chemical factors are shown in Table S2.” (Lines 328-330).

Table S2. Primary properties of topsoil (0-20 cm) at the test soil.

pH

Organic matter

(g kg −1)

Total nitrogen

(g kg −1)

Alkali-hydrolyzable nitrogen

(mg kg −1)

Available phosphorus

(mg kg −1)

Available potassium

(mg kg −1)

6.15

26.38

1.60

164.13

29.33

141.25

“The air-dried soil was sieved through a 4 mm mesh to remove any coarse material and vegetative matter” (Lines 330-331).

“The soil water treatment were performed as previously described by Xu et al. [27], with slight modifications. Different amounts of water were added to the bottom trays of the pots, and the wheat plants were subjected to two water treatments: well-watered conditions (WW, maintaining 80% of field capacity for the first 4 days, and then drying for 2 days) and moderate soil drying (SD, maintaining 45% of field capacity for the first 4 days, and then drying for 2 days)”. (Lines 339-344).

We have cited the reference in Materials and Methods: “27. Xu, F.; Liao, H.; Yang, J.; Zhang, Y.; Yu, P.; Cao, Y.; Fang, J.; Chen, S.; Li, L.; Sun, L.; et al. Auxin-producing bacteria promote barley rhizosheath formation. Nat. Commun. 2023, 14, 1–12, https://doi.org/10.1038/s41467-023-40916-4”.

Please see the revised manuscript for details (Lines 328-331; Lines 339-344).

Question #3: Regarding the RNA extraction protocol, I believe adding details about the method would help to better understand it.

Answer: Yes, we have added the relevant information: “Total RNA was extracted from roots using a RNAprep Pure Plant Plus Kit (DP441; TIANGEN, China) according to the manufacturerr’s instructions. First-strand cDNA was synthesized using a First Strand cDNA Synthesis kit (TransGen Biotech, China) according to the manufacturer’s instructions. Quantitative real-time PCR (qRT-PCR) assays were performed as described by Li et al. [39]. The expression levels were normalized to the expression of the wheat Actin-6 gene. The primers specific for each gene are listed in Table S3” (Lines 364-370). Please see the revised manuscript for details (Lines 364-370).

Question #4: I would like to see a conclusion with a pointer to future investigations.

Answer: Many thanks for your good comments. Yes, we have revised the conclusion as suggested: “This study comprehensively elucidated the responding mechanisms of soil drying on wheat root elongation based on physiological–biochemical performance and transcriptional analysis. RNA-Seq analysis indicated that the upregulated genes in the wheat roots under SD were mainly enriched in starch and sucrose metabolism pathways and the biosynthesis pathway of benzoxazinoids. Various plant hormone-related genes were also differentially expressed under SD. The concentrations of CZ, ABA, and IAA increased significantly under SD, whereas the SA, JA, SAG, and indole-3-formaldehyde (ICADd) concentrations significantly decreased under SD. ABA, JA, and ethylene affect the biosynthesis of plant secondary metabolites pathway under SD. These results suggest that changes in these metabolic pathways and plant hormones contribute to wheat root elongation to obtain water from deeper layers. In conclusion, this study investigated transcriptomes and hormones in wheat roots under soil drying stress and will be useful for clarifying the mechanism of drought stress in wheat”. Please see the revised manuscript for details (Lines 393-405).

Thanks again to the referee for your careful reviewing.

With Best Wishes,

Yours truly,

Jianhua Zhang and Nanyan Zhu

The Chinese University of Hong Kong, Hong Kong, China, E-mail: jzhang@hkbu.edu.hk;

College of Animal Science and Technology, Yangzhou University, Yangzhou, China, E-mail:dx120200148@stu.yzu.edu.cn.

Round 2

Reviewer 2 Report

Comments and Suggestions for Authors

 Dear authors,

I agree with the modifications you have made and have no further comments.